# Reversible structural evolution of sodium-rich rhombohedral Prussian blue for sodium-ion batteries

Wanlin Wang[1,6], Yong Gang[2,6], Zhe Hu[1], Zichao Yan [1], Weijie Li[1], Yongcheng Li[2], Qin-Fen Gu [3,4✉], Zhixing Wang[5], Shu-Lei Chou [1✉], Hua-Kun Liu [1] & Shi-Xue Dou [1]

Iron-based Prussian blue analogs are promising low-cost and easily prepared cathode materials for sodium-ion batteries. Their materials quality and electrochemical performance are heavily reliant on the precipitation process. Here we report a controllable precipitation method to synthesize high-performance Prussian blue for sodium-ion storage. Characterization of the nucleation and evolution processes of the highly crystalline Prussian blue microcubes reveals a rhombohedral structure that exhibits high initial Coulombic efficiency, excellent rate performance, and cycling properties. The phase transitions in the as-obtained material are investigated by synchrotron in situ powder X-ray diffraction, which shows highly reversible structural transformations between rhombohedral, cubic, and tetragonal structures upon sodium-ion (de)intercalations. Moreover, the Prussian blue material from a large-scale synthesis process shows stable cycling performance in a pouch full cell over 1000 times. We believe that this work could pave the way for the real application of Prussian blue materials in sodium-ion batteries.

[1] Institute for Superconducting and Electronic Materials, University of Wollongong, Innovation Campus, Squires Way, North Wollongong, NSW 2522, Australia. [2] Liaoning Starry Sky Sodium-ion Battery Co., Ltd., Laser industrial park, High-tech district, Anshan 114000 Liaoning, China. [3] Australian Synchrotron (ANSTO), 800 Blackburn Road, Clayton, VIC 3168, Australia. [4] Institute for Computational Materials Science, School of Physics and Electronics, Henan University, Kaifeng, China. [5] School of Metallurgy and Environment, Central South University, Changsha 410083 Hunan, China. [6] These authors contributed equally: Wanlin Wang, Yong Gang. ✉email: qinfeng@ansto.gov.au; shulei@uow.edu.au

O wing to the low-cost and natural abundance of sodium, sodium-ion batteries (SIBs) are considered as an alternative to lithium-ion batteries, which is expected to be utilized for gird-scale energy storage in the future[1–3]. Different cathode materials for SIBs include layered oxides[4,5], polyanionic compounds[6,7], and Prussian blue analogs (PBAs)[8,9], among them, PBAs have attracted tremendous attentions because their open framework structure could easily accommodate $Na^+$ and enable its fast transportation[10]. Compared with other cathodes, the high-temperature calcination is not required during synthesis of PBAs, which effectively lower the manufacturing costs[8]. These advantages make PBAs quite likely to be mass produced and widely used as low-cost cathodes material for SIBs in the future[11].

The generic formula of PBAs for SIBs could be represented as $Na_{2-x}M[Fe(CN)_6]_{1-y}\Box y \cdot nH_2O$, in which $x = 0–2$, M are usually single or multitransition metals (such as Fe, Mn, Co, Ni, etc), $\Box$ signify the vancancies occupied with coordinated water. Appealed by the cheapness of Mn and Fe resources, Mn-based PBAs ($Na_{2-x}MnFe(CN)_6$) was firstly reported as cathode for SIBs by Goodenough's group[12], they also found the electrochemical performances strongly depend on the phase evolution during cycling process, the pristine monoclinic phase of Mn-based PBAs could be converted to a rhombohedral structure after removing the interstitial water, and the cycling stability was improved due to higher reversibility of the rhombohedral phase[13,14]. However, the long cyclic stability is still unsatisfied due to its intrinsic poor electronic conductivity and the structrual distortion caused by Jahn-teller effect of $Mn^{3+}$[15,16]. In contrast, Fe-based Prussian white ($Na_{1.92}FeFe(CN)_6$) exhibits excellent cycling and rate performance, which was firstly reported by them as well, it demonstrates a rhombohedral phase due to an extreme high-sodium content and low-water content in structure[17]. However, the hydrothermal method they used is not ideal for practical application due to the low-yield from the single iron source and the toxic NaCN would exist as by-product[18]. Liu et al. firstly applied the environmentally friendly co-precipitation method to fabricate Fe-based PBAs but the cubic phase with low-sodium content was obtained, which shows nanoparticle with poor crystallinity and unstable cycle life <200 cycles, it is caused by numerous defects and coordinated water in structure due to the fast nucleation speed and precipitation process[19]. The irreversible phase transition for cubic $Na_{2-x}FeFe(CN)_6$ was found in our previous work by in situ powder X-ray diffraction (PXRD) analysis, the cubic structure with insufficient sodium content could not be restored after $Na^+$ extraction accompanied by the unit cell reduction[20]. Synthesizing sodium-rich $Na_{2-x}FeFe(CN)_6$ via scalable co-precipitation method has still been a challenge so far due to the rapid precipitation process, the small nano-size particle and irregular morhology with low-sodium content might hinder its real application due to difficulty in manufacturing and electrode coating process[21–23]. Therefore, it is of great importance to understand the particle nucleation and growth process during precipitation of $Na_{2-x}FeFe(CN)_6$ and its relationship to particle size, morphology, and crystalline structure, the phase transition of sodium-rich rhombohedral $Na_{2-x}FeFe(CN)_6$ during $Na^+$ extractions/insertions is needed to be investigated as well.

In this work, a series of sodium-rich $Na_{2-x}FeFe(CN)_6$ are fabricated via a controllable precipitation method and the evolution process of highly crystalline microcube with rhombohedral structure has been investigated. Synchrotron in situ PXRD shows that the sodium-rich $Na_{2-x}FeFe(CN)_6$ undergoes highly reversible three-phase transitions between rhombohedral, cubic and tetragonal upon $Na^+$ (de)intercalations. Excellent electrochemical performances are obtained in coin cells including high initial Coulombic efficiency, superior rate capability, and stable cycle stability. A large-scale synthesis of sodium-rich $Na_{2-x}FeFe(CN)_6$ is also demonstrated with a pouch full cell exhibiting stable cycling performance over 1000 times.

## Results

**Materials charaterizations.** A series of $Na_{2-x}FeFe(CN)_6$ samples were synthesized via a modified co-precipitation method under 25 °C in order to get the high-quality products in a simple way. Four samples were selected for representing the evolution of highly crystalline $Na_{2-x}FeFe(CN)_6$ and their PXRD patterns are displayed in Supplementary Fig. 1a, PB-S1 sample exhibits a cubic phase and the others are rhombohedral as the doublets in the marked area could be observed[23,24]. Rietveld refinement was employed to explore the crystallographic information of PB-S1 and PB-S3 samples, the details of structures are listed in Supplementary Tables 1 and 2, respectively. All PXRD data analysis was done in TOPAS 5 software. The PXRD data was first indexed to get unit cell, lattice parameters, and crystal symmetry information. Then the indexed unit cell was used for Le Bail fitting the PXRD data to derive the suitable peak profile, and lattice parameters. These derived data was fixed and used for further Rietveld refinement. The Rietveld refinement was done with initial structure models from ICDD PDF-4 2019 database. The occupancy of all atoms are allowed to refine to check the atomic occupancy with fixed APD values at B = 1. E.g. in PB-S1 sample in Supplementary Table 1, the free refined CN ligand is 1.081. As the results, we assumed and fixed the Fe and CN occupancy to be 100%. The structural water content may be disordered or without long range order. Another indication is if water is periodic located in the structure, space group Fm-3m maybe reduced. Na positions were calculated by difference electron maps, then Na occupancy was refined. Nevertheless, it is hard to locate water positions due to no periodic ordering of water positions across the unit cells (none constructive diffraction signal). As a result, PB-S1 sample exhibits a cubic phase with Fm-3m space group and $a = b = c = 10.3711(1)$ Å, however, the PB-S3 sample demonstrates a rhombohedral structure with R-3 space group with $a = b = 7.43079(1)$ Å, $c = 17.6133(1)$ Å $\alpha = \beta = 90°$, $\gamma = 120°$. The structural distortion of rhombohedral phase from cubic structure might be caused by higher sodium content in the structure. Inductively coupled plasma (ICP) and thermogravimetric analysis (TGA) were carried out to confirm the element concentration and water content for the four samples, and the results are illustrated in Supplementary Table 3 and Supplementary Fig. 1b, respectively. Combining the results, the chemical formula for PB-S1, S2, S3, and S4 samples could be represented as $Na_{1.53}Fe[Fe(CN)_6] \cdot 4.2H_2O$, $Na_{1.67}Fe[Fe(CN)_6] \cdot 3.9H_2O$, $Na_{1.73}Fe[Fe(CN)_6] \cdot 3.8H_2O$, and $Na_{1.68}Fe[Fe(CN)_6] \cdot 3.9H_2O$, respectively. However, the vancancies in samples are hard to be defined due to the limitation of ICP test, but sodium-rich $Na_{2-x}FeFe(CN)_6$ with rhombohedral phase with less coordinated water could be successfully obtained from the mild precipitation process. The refined PXRD patterns and structures of PB-S1 and PB-S3 as representative of cubic and rhombohedral phase are shown in Fig. 1.

Scanning electron microscope (SEM) images for PB-S1, PB-S2, PB-S3, and PB-S4 samples are shown in Fig. 2a–d, respectively, and the corresponding enlarged images are presented below in Fig. 2e–h, respectively. By carefully controlling various conditions during synthesis process, sodium citrate played the most important role in the precipitation process as it acts as the chelating agent and sodium supplement at the same time[25], which is favorable for retarding the precipitation speed and increasing the sodium content in $Na_{2-x}FeFe(CN)_6$. Gradually increasing the concentration of sodium citrate during precipitation process would be helpful for primary particle growing bigger

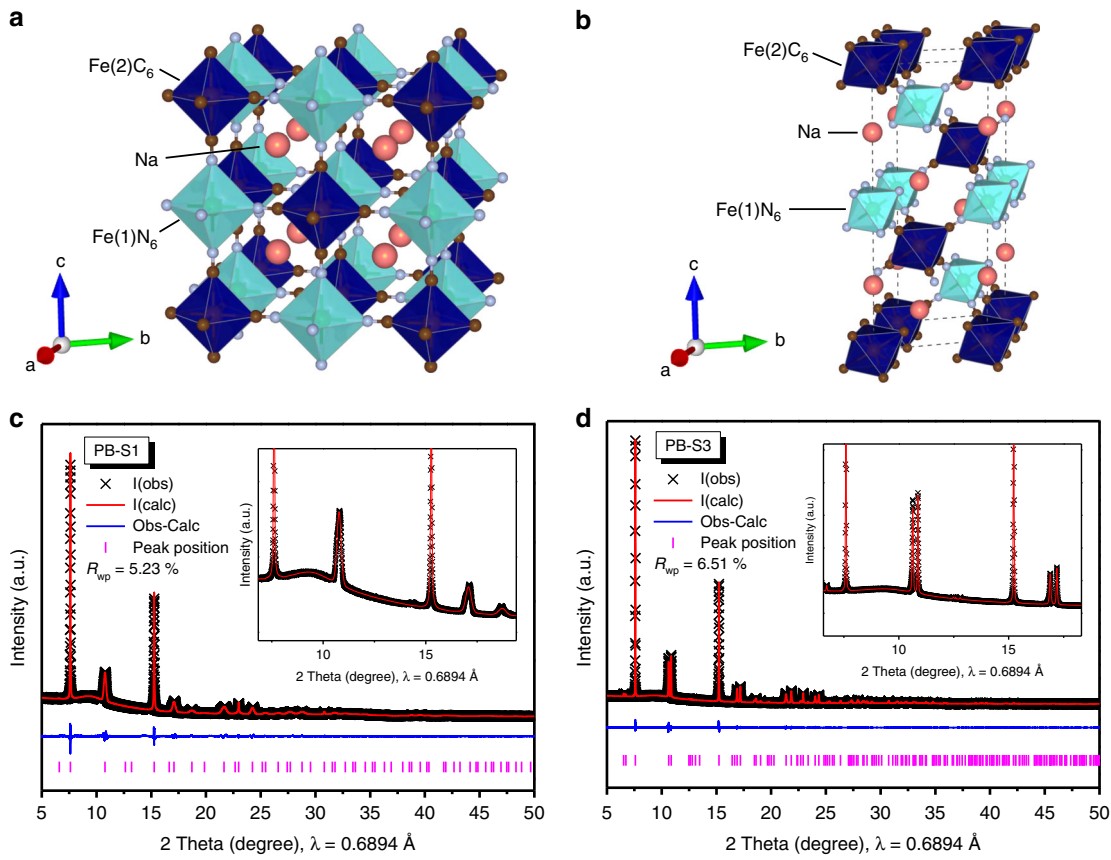

**Fig. 1 Structure and Rietveld refinement of cubic and rhombohedral samples. a**, **b** Schematic structure (Na in red, C in brown, N in grey, Fe1 and Fe2 in cyan and blue, respectively) and (**c**, **d**) Rietveld refinement synchrotron PXRD pattern of cubic PB-S1 and rhombohedral PB-S3 sample, respectively.

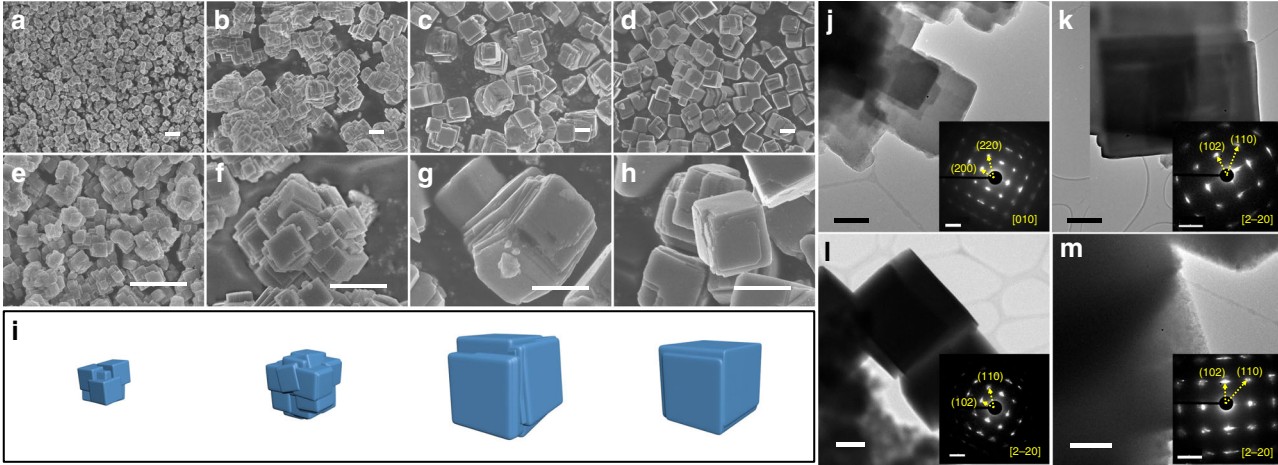

**Fig. 2 SEM, TEM with SAED of different samples.** SEM images of (**a**–**d**) PB-S1, PB-S2, PB-S3, and PB-S4 samples respectively and (**e**–**h**) corresponding enlarged images, (**i**) illustration of evolution process of highly crystalline $Na_{2-x}FeFe(CN)_6$, (**j**–**m**) STEM images with inset SAED of PB-S1, PB-S2, PB-S3, and PB-S4 samples respectively. Scale bars: 1 μm (**a**–**h**); 200 nm (**j**, **k**); 500 nm (**l**, **m**); 2 1/nm for inset SAED patterns (**j**–**m**).

and finally the single microcube was obtained. It is worth noting that the chelating agent is not the only factor that affects the precipitation process. For comparison, other conditions during co-precipitation process were investigated as well and their SEM images are presented in Supplementary Fig. 2. For example, no matter how many complex agent was used, the primary nanoparticle could not grow bigger if there is no $N_2$ used in precipitation process, as shown in Supplementary Fig. 2b, the oxidation of $Fe^{2+}$ would cause low-sodium content due to the valence equilibrium in $Na_{2-x}FeFe(CN)_6$ ($x > 0$). In addition, the

concentration of sodium citrate played an important role in particle size and morphology evolution, the particle size is extremely small with irregular morphology if only insufficient sodium citrate was used in precipitation process (Supplementary Fig. 2a). The chelating time between $Fe^{2+}$ and citrate would affect the morphology as well (Supplementary Fig. 2c), the primary particle demonstrates sub-micro meter, indicating the $Fe^{2+}$ could not be well-coordinated by citrate in a short time and the reaction between $Fe^{2+}$ and $[Fe(CN)_6]^{4-}$ was still too fast. Sodium citrate was added in both $FeSO_4$ and $Na_4Fe(CN)_6$ solution, which is

different than other reports and it is important for the slow nucleation, sodium citrate in $Na_4Fe(CN)_6$ solution is also helpful to slow down the precipitation speed between $Fe^{2+}$ and $[Fe(CN)_6]^{4-}$ because $Fe^{2+}$-citrate coordinated compound would be formed as a kind of competition. When sodium citrate was removed from $Na_4Fe(CN)_6$ solution, the morphology of as-obtained sample is presented in Supplementary Fig. 2d, which is quite similar with Supplementary Fig. 2c, the small primary particle indicates that the nucleation speed is still fast. The source of sodium salt was investigated as well, Supplementary Fig. 2e demonstrates when substituting sodium citrate in $Na_4Fe(CN)_6$ solution by equal mole NaCl, as many other paper reported[21,26,27]; however, we found that NaCl is unfavorable for particle growth and the irregular morphology was obtained as a result. It can be concluded that the atmosphere, chelating time between the complex agent and $Fe^{2+}$, the sodium concentration and even the source of sodium salt in different solutions would strongly affect the nucleation speed, particle size, and morphologies at the same time. By carefully controlling different precipitation conditions together, the evolution of highly crystalline particle could be seen from irregular nanoparticle (Supplementary Fig. 2a) to aggregated sub-microcubes (Fig. 2e, f) and finally to single microcube (Fig. 2g, h), part of process is illustrated in Fig. 2i. The uniform microcube particle for cathode material is ideal for electrode coating, increasing the volume density and alleviating the side reactions between electrode and electrolyte when it comes to real application, however, most of the reported PBAs have shown nanoparticles so far[16,19,21,22]. During the entire precipitation process, keeping the low valence of $Fe^{2+}$ is the key point to fabricating sodium-rich $Na_{2-x}FeFe(CN)_6$ but the X-ray photoelectron spectroscopy (XPS) results of final powder show that part of the $Fe^{2+}$ has been oxidized to $Fe^{3+}$ (Supplementary Fig. 4)[21]. Some digital photographs were taken during the experiment (Supplementary Fig. 5). The black and yellow color of the $FeSO_4$ solution demonstrates that $Fe^{2+}$ would be oxidized without $N_2$ protection (Supplementary Fig. 5a, b), the color change of slurry from white to blue is the evidence that $Fe^{2+}$ in $Na_{2-x}FeFe(CN)_6$ would be inevitably oxidized during washing and drying processes due to the part of $Na^+$ lost from $Na_{2-x}FeFe(CN)_6$ (Supplementary Fig. 5c–g)[17]. The evenly elemental distribution was detected by energy dispersive spectrometer (EDS), which is shown in Supplementary Fig. 3. Scanning transmission electron microscopy (STEM) images were carried out for confirming the phase of the four samples as well, displaying in Fig. 2j–m, respectively. Due to the electron beam sensitivity of the PBAs, only selected area electron diffraction (SAED) was applied to confirm the structures. The (200) and (220) planes along the [001] axis can be observed from cubic structure of PB-S1, the (102), (110), and planes along the [2–20] axis from rhombohedral phase could be seen for the PB-S2, PB-S3, and PB-S4 samples.

**Electrochemical properties investigations of cubic $Na_{1.53}Fe[Fe(CN)_6]\cdot4.2H_2O$ and rhombohedral $Na_{1.73}Fe[Fe(CN)_6]\cdot3.8H_2O$.** Due to different structure of cubic PB-S1 sample and rhombohedral PB-S2, S3, S4 samples, and PB-S3 sample contains the highest sodium concentration with the strongest PXRD intensity, PB-S1 ($Na_{1.53}Fe[Fe(CN)_6]\cdot4.2H_2O$) and PB-S3 ($Na_{1.73}Fe[Fe(CN)_6]\cdot3.8H_2O$) samples were selected to investigate the electrochemical performances as representative of the cubic and rhombohedral phase, respectively. Figure 3a shows the initial charge–discharge curves, with the discharge capacity for both sample around 116 mA h g$^{-1}$, but the charge capacity of PB-S1 is about 20 mA h g$^{-1}$ lower than that of PB-S3, demonstrating the initial Coulombic efficiencies (ICE) 120% and 97.4%, respectively,

the abnormal ICE beyond 100% for cathode materials indicates that part of the $Na^+$ from anode sodium plate was inserted into the cubic phase during the discharge process due to its sodium-poor phase, it would also be affected by oxidation of materials, which is also found from other reports[19,22,28,29]. The ideal ICE for a cathode material should be close or a slightly lower than 100%, which is extremely important when it comes to making full cell as the $Na^+$ only come from the cathode side. To get the higher ICE for $Na_{2-x}FeFe(CN)_6$, the sodium content should be high and the material should be protected from oxidization during synthesis. The plateau around 3 V of PB-S1 is much shorter than that of PB-S3, which can be attributed to the low-sodium content in cubic structure. The rate performances are presented in Fig. 3b, the capacity of PB-S3 could retain 83% at high-current density of 500 mA g$^{-1}$ based on capacity at 10 mA g$^{-1}$ compared with 58% for PB-S1, and even ~70 mA h g$^{-1}$ discharge capacity could be retained at high current density of 2000 mA g$^{-1}$ for PB-S3. In addition, the cycling stability was tested after activated coin cell with 10 mA g$^{-1}$ for two cycles and then cycled at 100 mA g$^{-1}$. The rhombohedral PB-S3 shows better cyclic stability as well (Fig. 3c), with the capacity maintained at 71% after 500 cycles for PB-S3 but it is only 66% for PB-S1 after 200 cycles. The charge–discharge curves at different cycles for both sample are displayed in Supplementary Fig. 6a, b. Cyclic voltammograms was employed for analyzing redox couples of $Fe^{2+}/Fe^{3+}$ from different sites in structure[19], the results for PB-S1 and PB-S3 samples are presented in Fig. 3d, e, respectively. The peaks at 2.8–3.1 V are related to the redox reactions of high spin $Fe^{2+}$ connected with N atoms, and peaks round ~3.8 V corresponds to the low spin $Fe^{2+}$ connected with C atoms[29], while the oxidization peak for PB-S1 over 3.8 V is more obvious than that of PB-S3, which is consistent with the higher long plateau in the charge curve of PB-S1 (Fig. 3a), the reason might due to the deficient $Fe^{2+}$ from N sites in the cubic phase, more $Na^+$ would be extracted during charge process by oxidization of $Fe^{2+}$ from C sites. While the small redox peaks around 3.3–3.4 V for both sample might be caused by $Na^+$ extracted/inserted from some other sites in the structure[19,30]. Galvanostatic intermittent titration technique (GITT) testing was employed to analyze the kinetic properties of $Na^+$ in two samples, as shown in Fig. 3f, g, respectively. The GITT test was conducted after four cycles of the fresh coin cell at current density of 10 mA g$^{-1}$, in which the cell was alternately charged for 10 min followed by 60 min resting, then discharged in the same way. For instance, a single GITT step of sample PB-S3 is presented in Supplementary Fig. 6c and the roughly linear relationship of $E$ vs. $\tau^{1/2}$ is presented in Supplementary Fig. 6d. The average calculated $Na^+$ diffusion co-efficients ($D_{Na^+}$) of PB-S1 and PB-S3 samples are quite close due to both open framework structure but $D_{Na^+}$ of PB-S3 sample is a little higher, which might due to more stable sodium-rich rhombohedral structure. The comparison of electrochemical performance of our PB-S3 sample with other reports of precipitated $Na_{2-x}FeFe(CN)_6$ samples is displayed in Supplementary Table 4, benefiting from the well-controlled mild precipitation process, PB-S3 sample demonstrates better performance especially ICE and long-cycling performance. However, the capacity of PB-S3 sample is still far from theoretical capacity of PBAs, which might due to defects and coordinated water in structure and the slightly oxidization of material.

**Investigation of phase transitions of cubic and rhombohedral structures during cycling.** Synchrotron in situ PXRD was performed to analyze the structural evolutions of the PB-S1 ($Na_{1.53}Fe[Fe(CN)_6]\cdot4.2H_2O$) and PB-S3 ($Na_{1.73}Fe[Fe(CN)_6]\cdot3.8H_2O$) samples with initial cubic or rhombohedral phase, respectively, during $Na^+$ extraction and insertion processes. Supplementary Fig. 7

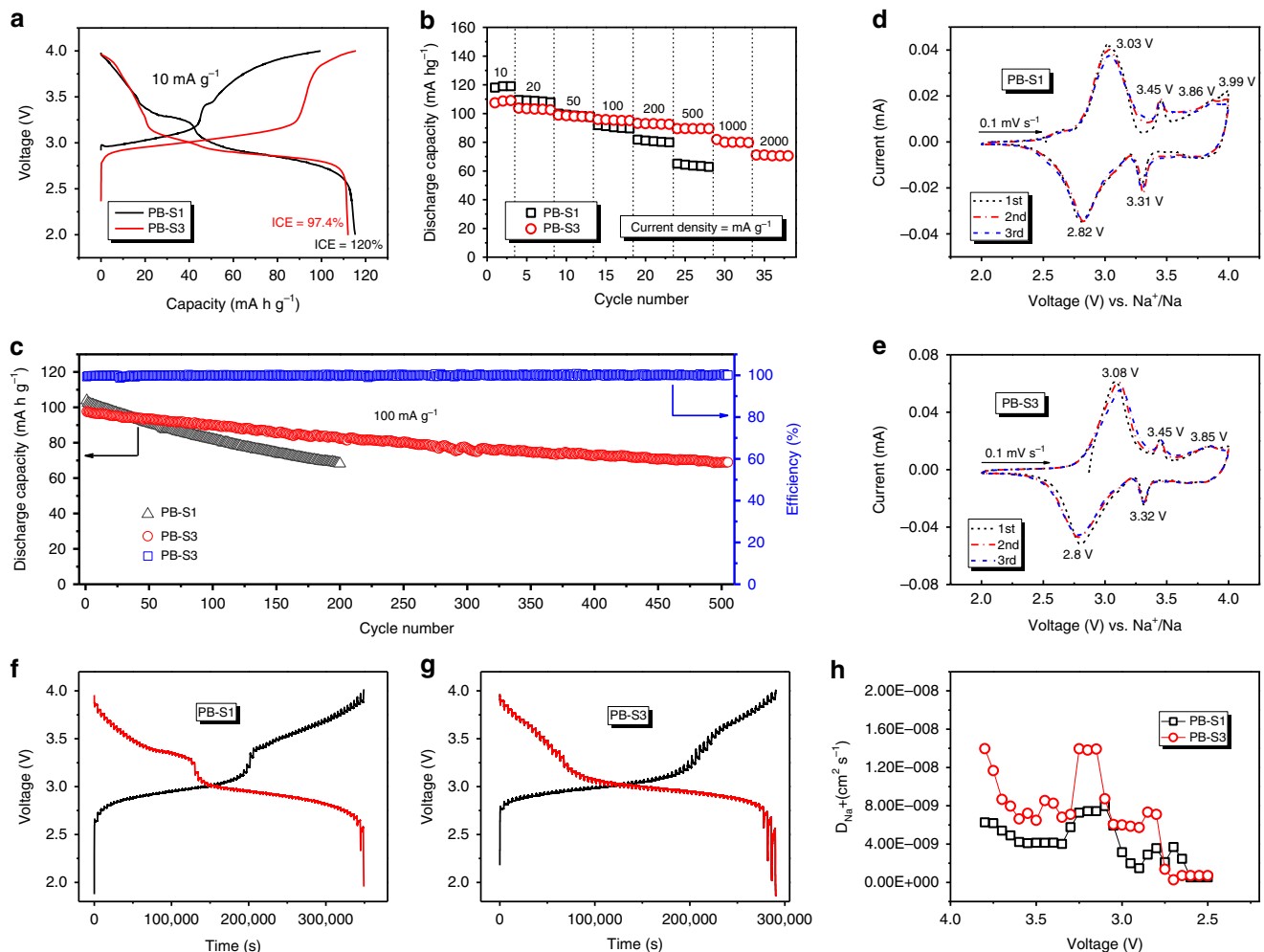

**Fig. 3 Electrochemical properties of cubic PB-S1 and rhombohedral PB-S3 samples.** The (**a**) initial charge–discharge curves, (**b**) rate and (**c**) cycling performance, (**d**, **e**) CV curves, (**f**, **g**) GITT curves, and (**h**) calculated $D_{Na^+}$ during the discharge process of the PB-S1 and PB-S3 samples.

shows the reflection planes of (200), (220), and (400) of in situ PXRD patterns of the initial cycle of cubic PB-S1 sample, all the peaks shifted toward higher angles during charge process but they could not return to original positions after discharging, indicating the irreversible lattice shrink occurred after Na$^+$ extraction, the result is consistent with our previous work[20]. Two peaks were observed at the end of discharging process of (200) and (400) reflections, which might due to part of Na$^+$ from anode side inserting into cathode side and a mixture of cubic and rhombohedral phase was formed. In comparison, the first two cycles were recorded for rhombohedral PB-S3 sample, three representative reflection of planes (012), (110)/(104), and (024) are illustrated in Fig. 4c. The peak shift for all peaks is highly reversible in two cycles, indicating the rhombohedral framework could be well restored after Na$^+$ extraction/insertions. The peak broadening of (110)/(104) might be caused by a slightly changed cell orientation or internal strain in the cell cathode during the measurement. The PXRD patterns of electrode of pristine and discharged to 2.0 V state were confirmed by laboratory PXRD facility as well, and clearly a rhombohedral phase could be observed from double peaks around 24° (Fig. 5c). In terms of obvious peaks of (012) and (024) in synchrotron in situ PXRD data (Fig. 4c), during the first cycle, the peaks shifted towards higher angles at the onset of charging process, which demonstrate that the lattice parameters were reduced after Na$^+$ extracted from the structure. After charged to 3.2 V, the (110) and (104) peaks merged together with the phase

transition from the rhombohedral to the cubic structure. Furthermore, the peak shifting stopped around 3.2 V and then slightly shifted back to lower angles when keep charging over 3.2 V until 4.0 V. This was caused by the phase transition from a cubic to a tetragonal phase (P4/mmm space group) due to more Na$^+$ being extracted and the lattice was expanded[13]. For comparison, the simulated PXRD data from three phases are illustrated in Fig. 5b, which is consistent with our selected in situ synchrotron PXRD results (Fig. 5a). The three-phase transitions from rhombohedral, cubic, and tetragonal for PB-S3 sample were also confirmed by selected Le Bail fitted PXRD patterns from pristine, 3.2 and 4.0 V charging states, as shown in Fig. 5d, e, f, respectively. The discharge process of the first cycle demonstrates that the PXRD patterns changed in reversed order compared with charging process, and all peaks finally back to their original positions, manifesting the phase converted back from tetragonal to cubic to rhombohedral again. The second cycle shows almost the identical phenomenon, indicating the excellent reversibility of the rhombohedral structure. Figure 4a shows the 2D contour map of (012) reflection and the peak shift can be easily observed. The normalized volume variation in a primitive unit cell is shown in Fig. 4b, the rhombohedral structure experienced both shrink and expansion with small-volume variation ~4% during single-charge or discharge process and the structure was restored as well. This three-phase structural evolutions of Na$_{2-x}$FeFe(CN)$_6$ is different than other reports due to high sodium content in our PB-S3 sample[22,31,32], the reversible

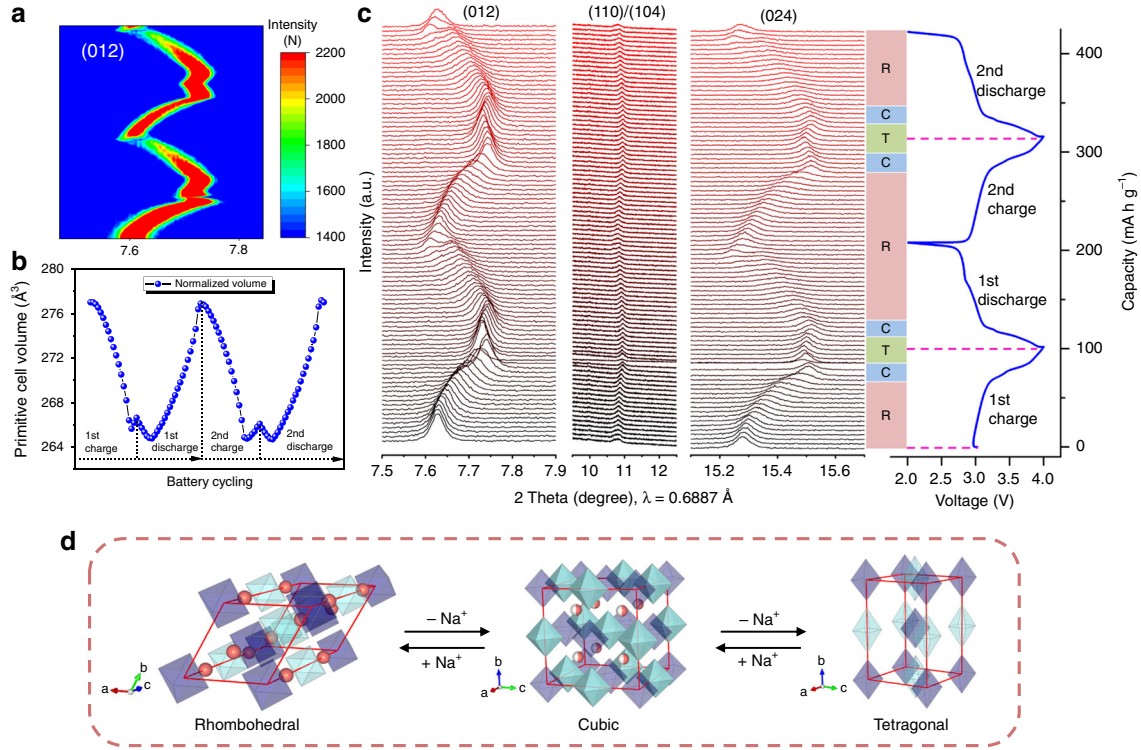

**Fig. 4 Investigation of phase transitions of Na$_{1.73}$Fe[Fe(CN)$_6$]·3.8H$_2$O sample with rhombohedral structure during cycling. a** 2D contour plot of (012) reflection plane, **b** normalized volume during charge–discharge process obtained from synchrotron in situ PXRD patterns of rhombohedral PB-S3, **c** (012), (110)/(104), and (024) reflection planes of synchrotron in situ PXRD patterns, **d** schematic three-phase evolutions during cycling of rhombohedral PB-S3.

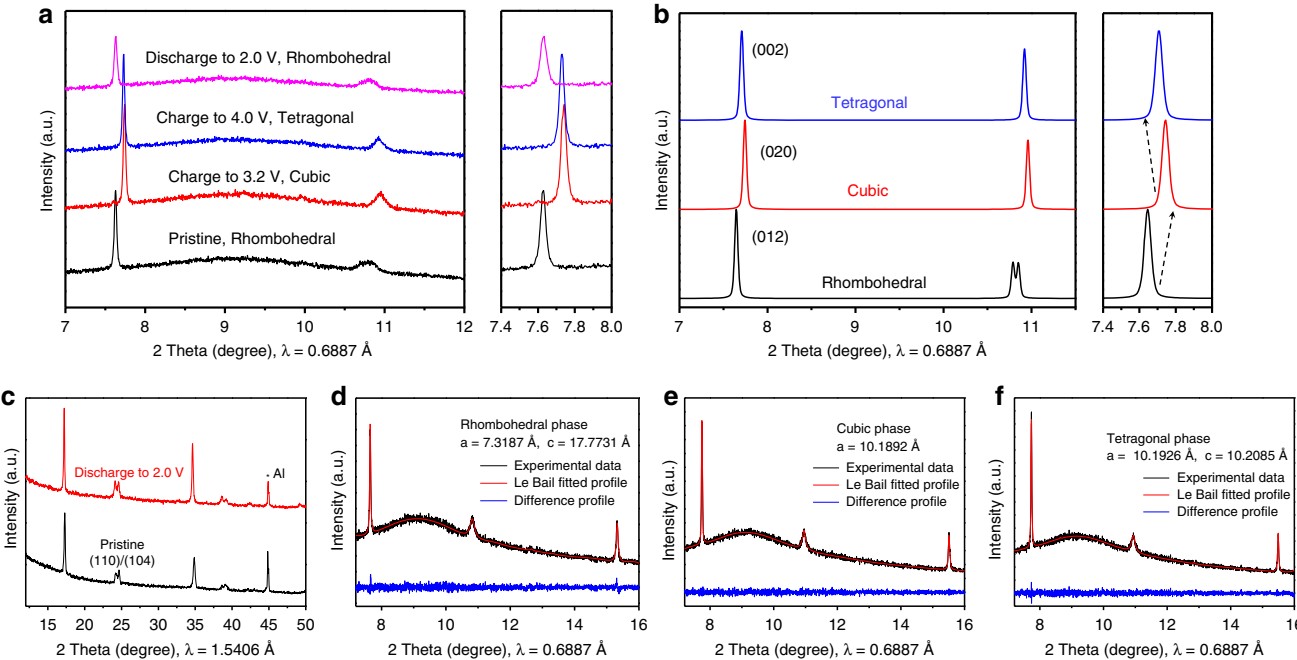

**Fig. 5 Structural characterization of PB-S3 sample during cycling. a** Synchrotron PXRD patterns of PB-S3 sample at different voltages, (**b**) simulated PXRD patterns of three different phases, (**c**) PXRD patterns of electrode of pristine and 2.0 V discharging state, (**d–f**) Le Bail fitted PXRD patterns from pristine PB-S3 (rhombohedral), 3.2 V charging (cubic), and 4.0 V charing states (tetragonal).

three-phase transitions for rhombohedral Na$_{2-x}$FeFe(CN)$_6$ were observed for the first time. Moreover, the phase transitions for rhombohedral structure is consistent with the charge–discharge curve of PB-S3 as well (Fig. 3a). The rhombohedral to cubic structure corresponds to the long plateau below 3.2 V, which

contributes majority of the capacity with stable high spin Fe$^{2+}$ redox reaction, and the cubic to the tetragonal phase transition could account for the extra charge capacity more than that of cubic phase with insufficient sodium content. The highly reversible phase transitions for rhombohedral structure of PB-S3 could also explain

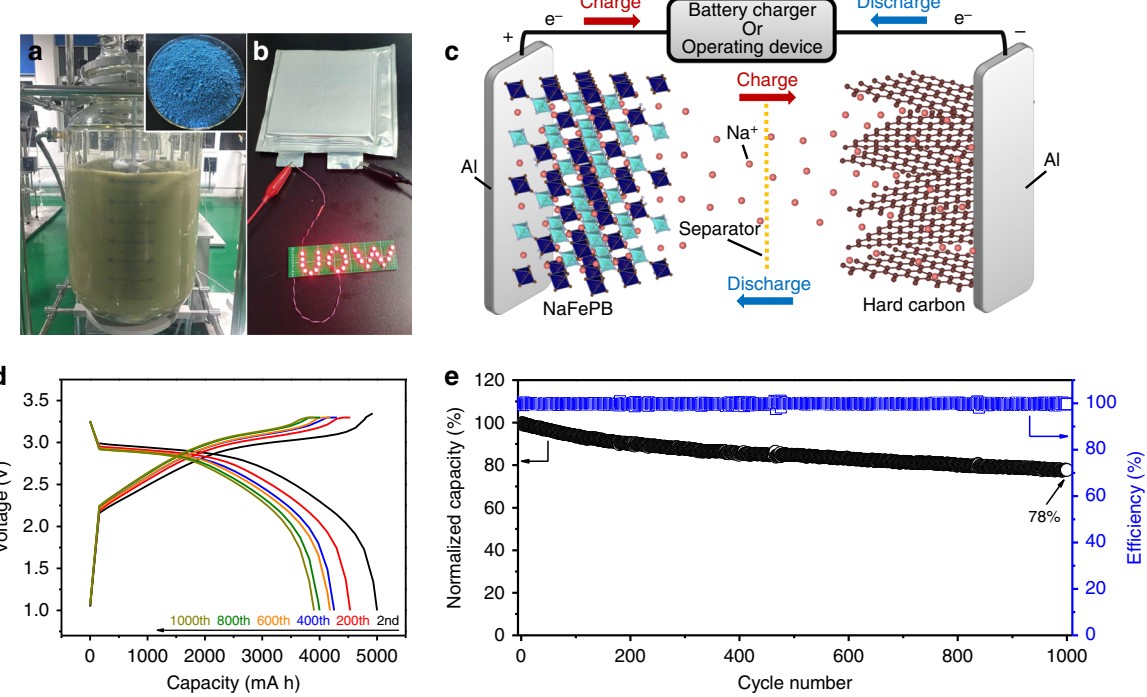

**Fig. 6 Demonstration of large-scale production of Na$_{2-x}$FeFe(CN)$_6$ and pouch full cell performance. a** Digital image of synthesis of Prussian white Na$_{2-x}$FeFe(CN)$_6$ in 100 L reactor and powder of final product, (**b**) digital image of pouch full cell connected with red LED lights, (**c**) working mechanism of pouch full cell, (**d**) charge-discharge curves of pouch full cell, (**e**) and cycling performance of pouch full cell.

for its high initial Coulombic efficiency (97.4%), superior rate capability, and stable cycling performance as well. Finally, the three-phase evolutions during charge–discharge process of rhombohedral Na$_{2-x}$FeFe(CN)$_6$ is schematically presented in Fig. 4d.

**Electrochemical properties of pouch full cell**. To demonstrate the feasibility of practical application of rhombohedral Na$_{2-x}$FeFe(CN)$_6$, a scale-up synthesis was conducted and a pouch full cell was fabricated as well. Figure 6a demonstrates a large-scale precipitation experiment by using a 100 L reactor with 5 kg yield, which is based on the expansion of precipice of PB-S3 sample, the precipitation process were carefully controlled and the sodium-rich prussian white was sucessfully obtained, the digital image of some final product is attached in Fig. 6a. Supplementary Fig. 8a, b present the PXRD pattern and SEM image of the final product, respectively, which also demonstrate the rhombohedral strucutre with microcube morphology. A pouch full cell was assembled by using scalable fabricated PBAs as cathode and commercial hard carbon was used as anode counterpart. Figure 6b shows the digital image of pouch cell connected with red LED lights and the working mechanism of pouch full cell is illustrated in Fig. 6c. Long-cycling performance over a thousand times for pouch cell has been tested, the test voltage range is set between 1.0 and 3.2 V in order obtain the stable cycling performance. Figure 6d shows charge–discharge curves of pouch cell at different cycles under a current density of 1 C, obvious discharge plateaus around 2.9 V are observed. Corresponding cycling performance of the pouch cell is displayed in Fig. 6e, benefiting from the highly stable rhombohedral Na$_{2-x}$FeFe(CN)$_6$ cathode material, the capacity retention was maintained at 78% over 1000 cycles.

We have successfully synthesized sodium-rich Na$_{2-x}$FeFe (CN)$_6$ with highly reversible rhombohedral structure via a simple scalable co-precipitation method, by carefully controlling the precipitation process, the nucleation and growth process from nanoparticle to highly crystalline Na$_{2-x}$FeFe(CN)$_6$ microcube is discovered. Synchrotron in situ PXRD shows that the rhombohedral structure is highly reversible upon Na$^+$ extractions/insertions with three-phase transitions between rhombohedral, cubic, and tetragonal. Benefiting from its stable structure, Na$_{1.73}$Fe[Fe(CN)$_6$]·3.8H$_2$O sample shows excellent electrochemical performance with high initial Coulombic efficiency 97.4%, 70 mA h g$^{-1}$ discharge capacity was retained at current density of 2000 mA g$^{-1}$, capacity retention was maintained at 71% after 500 cycles. To demonstrate the practical application of rhombohedral Na$_{2-x}$FeFe(CN)$_6$, a scale-up precipitation experiment was conducted by using a 100 L reactor and a pouch full cell was fabricated as well, which shows a stable cycling performance over a thousand times. Our work might guide rational synthesis for other PBAs from scalable precipitation method, which would pave the way for mass producing PBAs and designing high-performance SIBs in the future.

## Methods

**Synthesis of Na$_{2-x}$FeFe(CN)$_6$ samples**. All chemicals for synthesizing PBAs in laboratory were purchased from Australian Sigma Aldrich, and Na$_{2-x}$FeFe(CN)$_6$ samples were synthesized via a modified precipitation methods at 25 °C. For example, for a sample labeled as PB-S1: 2.81 g sodium citrate and 1.67 g FeSO$_4$·7H$_2$O were mixed in deionized water and a 100 ml solution was prepared, labeled as solution A. Then 2.81 g sodium citrate and 1.96 g Na$_4$Fe(CN)$_6$·10H$_2$O were dissolved in another 100 ml solution, labeled as solution B. Both of solutions were bubbled with N$_2$ to protect Fe$^{2+}$ from oxidation. Solution A was stirred for 3 h before using. Solutions A and B were transferred quickly into a constant pressure drop funnel and a three-neck flask (under N$_2$), respectively. Then, solution A was added dropwise into solution B, and the mixture was kept stirring at 800 rpm for 6 h and aged for few hours before centrifugation, the precipitate was washed with deionized water and ethanol three times, respectively, and finally the wet powder was dried in vacuum oven at 120 °C for 12 h. Other samples such as PB-S2, PB-S3, and PB-S4 were synthesized by the same route except that the sodium citrate in both solutions A and B were 3.75, 7.5, and 15 g, respectively. To understand the other factors that affect the precipitation process, some other samples were synthesized for comparison as well. Sample PB-SA was fabricated

based on recipe of for PB-S1 but only half the amount of sodium citrate was used in both A and B solutions. The other samples are all based on the recipe and preparation route of sample PB-S3; Sample PB-SB was synthesized without $N_2$ during the whole experiment; Sample PB-SC was synthesized by reducing the stirring time of solution A to 30 min; Sample PB-SD was synthesized by removing the sodium citrate from solution B; Sample PB-SF was synthesized by substituting the sodium citrate in solution B by an equal amount of NaCl in the same mole ratio. All chemicals for producing PBAs in the 100 L reactor were purchased from Annaizhi (Tianjin) Technology Co., Ltd., China, and the hard carbon was purchased from KURARAY Co., Ltd., Japan.

**Materials characterization**. The crystalline structures of the as-prepared samples were examined by laboratory PXRD on a GBC MMA diffractometer with a Cu Kα source. The powder in capillaries and batteries were also examined in situ with PXRD method on the powder diffraction beamline at the Australian Synchrotron at a wavelength (λ) of 0.6894 and 0.6887 Å respectively, calibrated with the standard reference material ($LaB_6$ 660b). The obtained powder data from the synchrotron were indexed and refined with TOPAS 5 (Bruker) software. The Morphologies of the as-prepared samples were observed with a field emission SEM (JEOL JSM-7500FA). The element distribution was detected by EDS (JEOL JSM-6490), and more information was obtained with scanning transmission electron microscopy (STEM, JEM-ARM 200F), equipped with selected area electron diffraction (SAED). A Mettler-Toledo TGA/differential scanning calorimetry (TGA/DSC) STARe system was used to determine the water content in the sample with a program running from 50 to 500 °C ramped at 10 °C min$^{-1}$ in Ar. XPS (PHI5600, PerkineElmer) measurements were performed to obtain the valence information on Fe in the as-prepared samples. The concentrations of Na and Fe in the samples were measured by ICP analysis (OPTIMA 8000DV Optical Emission Spectrometers).

**Electrochemical measurements**. The electrode slurry for coin cell testing was mixed in a ratio of 70 wt% active materials, 20 wt% carbon black (super P, C45), 10 wt% polyvinylidene fluoride (PVDF) and appropriate amount of N-methyl pyrrolidone (NMP), the slurry was coated on aluminum foil and dried at 120 °C in a vacuum oven for 12 h. Then, the electrode was pressed under a pressure of 20 MPa and then punched into a round shape (0.95 cm in diameter). Sodium metal was cut and pressed to form the foil to be used as counter electrode. The electrolyte contains 1 M $NaClO_4$ dissolved in ethylene carbonate diethyl carbonate and propylene carbonate (EC: PC = 1:1) with 3% fluoroethylene carbonate (FEC) as additive (by volume), and glass fiber was used as separator. A glove box filled with Ar was used for assembling the coin cells (type 2032) with the concentration of $O_2$ and $H_2O$ lower than 0.1 ppm. The electrochemical data on the coin cells was obtained from a Neware battery system (Neware Technology Ltd.) by testing at different current densities within the voltage range from 2–4 V, the cycling performance was activated at current density 10 mA g$^{-1}$ three times and then cycled at 100 mA g$^{-1}$. Rate performance was charged at 10 and 20 mA for the first three and five cycles, and the later cycles were charged at both 50 mA g$^{-1}$, the discharge process was tested at gradually increasing current densities. Cyclic voltammetry tests were conducted by a Biologic VMP-3 electrochemical workstation between 2-4 V at a scan rate of 0.1 mV s$^{-1}$. A Land battery system was applied to perform GITT testing after four cycles of the fresh coin cell at current density of 10 mA g$^{-1}$ from 2-4 V, in which the cell was alternately charged for 10 min followed by 60 min resting, then discharged in the same way. The sodium-ion diffusion coefficient ($D_{Na^+}$) was calculated by using the slope taken from the linear range using the following simplified Eq. (1):

$$D_{Na^+} = \frac{4}{\pi\tau}\left(\frac{m_B V_m}{M_B S}\right)^2 \left(\frac{\Delta E_s}{\Delta E_\tau}\right)^2 \left(\tau \ll \frac{L^2}{D_{Na^+}}\right) \quad (1)$$

Where $m_B$, $Vm$, and $M_B$ represent the active mass, the molar volume, and the molar mass of the $Na_{2-x}FeFe(CN)_6$ material, respectively. $\tau$ signifies the duration of a current pulse, and $S$ is the geometric surface of the $Na_{2-x}FeFe(CN)_6$ electrode. $\Delta E_t$ represents the voltage gap between the two steady-state voltages in a single-step GITT process, and $\Delta E_t$ is the voltage variation during a current pulse. The in situ battery for synchrotron testing was modified from a 2032 coin cell at current density 20 mA g$^{-1}$ from 2-4 V, the data was collected by in situ synchrotron X-ray diffraction measurement by combined using a Neware battery system with an in-house designed setup, the data was analyzed via PDViPeR software. The capacity of pouch full cell was designed at 7 Ah and the capacity of anode is designed for 20% (by capacity) excess than that of cathode. Electrode slurry of both cathode and anode materials for pouch full cell were mixed in a ratio of 90 wt% active materials, 5 wt% carbon black (super P, C45), 5 wt% PVDF and appropriate amount of NMP the slurry was coated on aluminum foil and dried at 120 °C. 1 M $NaPF_6$ dissolved in ethylene carbonate diethyl carbonate and diethyl carbonate (EC: DEC = 1:1) with 2% FEC as additive (by volume), and polypropylene film was used as separator.

## Data availability
Data supporting the findings of this study are available from the authors on reasonable request. See author contributions for specific data sets.

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

## Acknowledgements

The authors are grateful for the Australian Renewable Energy Agency (ARENA) Project (G00849) and Liaoning Province Innovation Team Funding, W.W. is also supported from the China Scholarship Council (No. 201606370024). The authors thank Dr. Tania Silver for critical reading of the manuscirpt. Part of experiment was performed at PD beamline, Australian Synchrotron (ANSTO).

## Author contributions

W.W. and Y.G. contributed equally to this work. W.W., Y.G., Z.H., Z.Y., W.L. and S.C. designed and conducted the experimental work. Y.G. and Y.L. fabricated the pouch full cell and tested the electrochemical performance. Z.H. helped with material synthesis designing. Z.Y. and W.L. helped with analyzing the electrochemical data. Q.G. analyzed the synchrotron powder and in situ battery data. Z.W. performed the XPS and TGA analysis, S.C., H.L., Q.G. and S.D. helped with paper draft.

## Competing interests

The authors declare no competing interests.
