## [Peer Review File · Nature Communications]

Reviewers' comments:

Reviewer #1 (Remarks to the Author):

In this work Wang and coworkers synthesized Prussian White using a citrate-assisted method under a range of conditions. Electrochemical performance data of selected samples are presented. In operando XRD shows structural transformations of the material during cycling.

Using chelating agents in Prussian blue analogue synthesis, the most common being citrate, is a known method to increase particle size and reduce hexacyanometallate vacancy and water content. Previously this method has been applied to CoHCF_e, MnHCF_e and NiHCF_e. This is the first report for iron hexacyanoferrate.

Hexacyanometallate vacancy content is an important parameter in PBA performance. However, there is no reliable method to calculate the vacancy content for iron hexacyanoferrate. Moreover, coordinated water content is still high and the specific capacity is low, suggesting the materials still have high vacancy content.

One of the primary claims of the paper is evidence of structural evolution during cycling between rhombohedral – cubic – tetragonal phases. The in-operando data does not support this. In S8 there is no clear tetragonal phase. The authors cite data from manganese hexacyanoferrate which does transition to a tetragonal phase when fully oxidized but this is driven by a Jahn-Teller distortion in the Mn³⁺. For iron hexacyanoferrate a tetragonal transition is not expected and is not observed. There is some previous ex situ XRD showing this (Energy Environ. Sci., 2014, 7, 1643).

Can the authors explain how the Rietveld refinement was done? Was the occupancy of Fe₂ C N allowed to relax and what was the treatment of water within the sample (about 16% by mass from TGA)?

First cycle coulombic efficiency cannot be used for comparison as the material is oxidized by varying degrees.

Why do the authors think the specific capacity is lower than both the theoretical value and other values reported for iron hexacyanoferrate? The material is not better performing compared to others reported in terms of specific capacity and capacity retention.

Reviewer #2 (Remarks to the Author):

This manuscript reported a scalable synthesize method of sodium-rich Fe-based Prussian white for sodium ion storage. The nucleation and precipitation process are presented. Moreover, the phase transition mechanism during the sodium storage process is also studied. However, as the authors mentioned in the introduction part, the as-proposed cathode has been reported elsewhere, which also demonstrated excellent cycling and rate performance. There is no breakthrough in terms of electrochemical performance in this cathode. Besides, the sophisticated phase transition in this material does not benefit long-term cycling stability in practical application. On the other hand, the co-precipitation method is widely used in the preparation of PBAs. Hence, the novelty of this paper is not high enough, especially for Nature Communications. It is more suitable for a specific energy storage related journal. I cannot recommend its publication. There are some other questions for the authors' attention.

1. Why do the authors choose Sample 1 and Sample 3 to compare? How's the electrochemical

performance of Sample 2 and Sample 4?

2. Since the authors claimed that the ICE of the PB-S3 sample is beneficial for making full cell, it is more convincing to configure full cell to prove this.

3. How's the rate performance beyond 500 mA g⁻¹? The open framework of PBA is supposed to allow fast ion transportation.

4. There is no significant difference in the diffusion coefficient (Figure 3h), making the statement "which could explain the superior cycling and rate performance of the high sodium content rhombohedral structure" a little weak. Moreover, the cycling stability is not only related to the kinetic properties. The discussion here is too general.

5. The three-phase structural evolution should be further elucidated by Rietveld refinement.

Reviewer #3 (Remarks to the Author):

In this paper, the authors investigated Na_{2-x}FeFe(CN)₆ Prussian blue analogues as a kind of low-cost cathode material in sodium-ion battery. The precipitation process during synthesis was systematically studied and some interesting results were found, including particle size, morphology and the relationship to structure and battery performance. The rate and cycling performance have been improved by obtaining the high crystalline microcube particle with sodium-rich rhombohedral phase. In-situ XRD analysis shows a reversible three phase transitions for rhombohedral Na_{2-x}FeFe(CN)₆, which seems like a new mechanism of structural evolution for this material and has not been found in any other reports so far. The results are interesting and attractive which should be published here. But some minor revisions and questions should be addressed in the manuscript as well.

Synthesis methods were reported in other papers by using sodium-citrate as chelating agent like J. Mater. Chem. A, 2016, 4, 6036; Nano Energy 39 (2017) 273–283, what is the advantage between author's synthesis and others work?

The evolution of morphologies of four samples in Fig. 2 is needed to be explained, why would the primary particle aggregate and the size of PB-S4 sample become smaller than that of PB-S3 with increasing sodium citrate concentration.

In the GITT test in electrochemical performance, the "τ" is missing from the calculation formula of DNa⁺ in the Supporting Information, the GITT results needed to be verified.

There is a difference between unit cell volume of rhombohedral PB-S3 sample in Figure 4 and its XRD refined volume, could author explain it.

There should be more explanation for XRD stimulation in different three phases in Figure S8 in the supporting information. How did author confirm the tetragonal phase was formed.

Reviewers' comments:

Reviewer #1 (Remarks to the Author):

In this work Wang and coworkers synthesized Prussian White using a citrate-assisted method under a range of conditions. Electrochemical performance data of selected samples are presented. In operando XRD shows structural transformations of the material during cycling.

Using chelating agents in Prussian blue analogue synthesis, the most common being citrate, is a known method to increase particle size and reduce hexacyanometallate vacancy and water content. Previously this method has been applied to CoHCF, MnHCF and NiHCF. This is the first report for iron hexacyanoferrate.

Hexacyanometallate vacancy content is an important parameter in PBA performance. However, there is no reliable method to calculate the vacancy content for iron hexacyanoferrate. Moreover, coordinated water content is still high and the specific capacity is low, suggesting the materials still have high vacancy content.

In our manuscript, the content of Na and Fe elements for our samples was confirmed by inductively coupled plasma (ICP) analysis (OPTIMA 8000DV Optical Emission Spectrometers). However, C or N element could not be detected through ICP as this instrument could only test metal elements. We haven't got access to elements analysis facility for C and N yet, therefore, the vacancies analysis would be difficult for our samples. However, in reference "J. Mater. Chem. A, 2016, 4,6036–6041", $\text{Na}_{1.54}\text{Fe}[\text{Fe}(\text{CN})_6]_{0.96}$ with vacancy 0.04 was reported by elemental analysis. In another reference, "Nano Energy 39 (2017) 273–283", $\text{Na}_{1.63}\text{Fe}[\text{Fe}(\text{CN})_6]_{0.87}$ with vacancy 0.13 was reported via just ICP test, we have no idea how to get the CN content by ICP test, but the sodium content in our PB-S3 sample is higher than 1.7, hence, the vacancies in our sample might not that high, and the vacancies for our rhombohedral PB-S3 sample has been reduced compared to our PB-S1 sample, which shows a sodium-poor cubic phase with more crystalline water. The vacancies in $\text{Na}_{2-x}\text{Fe}[\text{Fe}(\text{CN})_6]$ were formed during precipitation process, and the fast precipitation process would introduce more vacancies in structure. Hence, in our work, we systematically studied all possible conditions that affected precipitation process, trying to slow down reaction speed to decrease the vacancies and crystalline water. The nucleation and particle growth process for high crystallinity $\text{Na}_{2-x}\text{Fe}[\text{Fe}(\text{CN})_6]$ was found for the first time, which has not been reported by others yet. It is important to understand the precipitation process for its practical scale-up application. An example of scale-up

synthesis of $\text{Na}_{2-x}\text{FeFe}(\text{CN})_6$ by using a large scale reactor (100 L) is shown in Figure 5 and stable pouch cell performance over a thousand times is demonstrated as well.

The specific capacity of our PB-S3 sample is 116 mA h g^{-1} , as far as we know, it is a very good value rather than low capacity, as other reported $\text{Na}_{2-x}\text{FeFe}(\text{CN})_6$ samples synthesized from co-precipitated methods are basically showing capacity around or less than 120 mA h g^{-1} , some of their initial discharge capacity is even higher than charge capacity, which is abnormal due to vacancies or oxidization of material. (J. Mater. Chem. A, 2018, 6, 8947). Hence, the discharge capacity should not be the only standard to define the quality of PBA. The charge capacity and initial Coulombic efficiency (ICE) are also needed to be taken into consideration. The ICE of our sample is 97.4 %, which is higher than almost all other references, which can be ascribed to the high sodium content in our sample and we endeavoured to prevent the material from oxidization during entire synthesis. We do admit the specific capacity is still far from theoretical capacity 170 mA h g^{-1} and even less than specific capacities of $\text{Na}_{2-x}\text{CoFe}(\text{CN})_6$ or $\text{Na}_{2-x}\text{MnFe}(\text{CN})_6$. There are some reasons behind it. First, the co-precipitation method is not as mild as hydrothermal method, which could reach high capacity 160 mAh g^{-1} , but hydrothermal method could not be used in real application as its limited yield and toxic NaCN might exist as a by-product. The faster reaction speed of co-precipitation would cause more vacancies in structure and lower the sodium concentration in sample. Second, the synthesizing process of $\text{Na}_{2-x}\text{FeFe}(\text{CN})_6$ was difficult to be controlled as Fe^{2+} would easily be oxidized during solution preparation, washing and even drying processes. When Fe^{2+} become Fe^{3+} , the sodium would be lost from structure in order to keep the valence balance in $\text{Na}_{2-x}\text{FeFe}(\text{CN})_6$ ($2-x < 2$), causing the lower capacity as well. However, Co^{2+} and Mn^{2+} are stable and would not be oxidized easily, therefore, more sodium would be kept in structure to get higher capacity. However, Cobalt is expensive, and Mn^{3+} would cause Jahn-teller effect and normally Nickel doping is used to stabilize structure, hence the cost would increase as well due to consumption of Nickel, this is why we chose Fe-based PBA as a cheap and relatively stable investigated target and no any other doping or coating modifications was adopted for our sample in order to keep the low cost of fabricating process. The reported $\text{Na}_{2-x}\text{FeFe}(\text{CN})_6$ from scalable co-precipitation method normally shows poor crystallinity cubic phase with small nano particles, unstable cycling performance was observed as well, which means the precipitation process was not carefully controlled. In our experiments, we have tried our best to carefully control the precipitation process of $\text{Na}_{2-x}\text{FeFe}(\text{CN})_6$ to get high quality products as much as we can. The high crystalline $\text{Na}_{2-x}\text{FeFe}(\text{CN})_6$ with high sodium content, micron size particle morphology, and rhombohedral structure was successfully obtained through a simple room temperature scalable method, which can not be found in any other references. Based on our experiences on how to control co-precipitation process, a scale-up production for $\text{Na}_{2-x}\text{FeFe}(\text{CN})_6$ is illustrated for the first time and the pouch full cell performance is demonstrated in Figure 5.

Figure 5 (a) Digital image of synthesis of Prussian white $\text{Na}_{2-x}\text{Fe}(\text{CN})_6$ in 100 L reactor and powder of final product, (b) digital images of pouch full cell connected with red LED lights, (c) working mechanism of pouch full cell, (d) charge-discharge curves of pouch full cell, (e) cycling performance of pouch full cell.

One of the primary claims of the paper is evidence of structural evolution during cycling between rhombohedral – cubic – tetragonal phases. The in-operando data does not support this. In S8 there is no clear tetragonal phase. The authors cite data from manganese hexacyanoferrate which does transition to a tetragonal phase when fully oxidized but this is driven by a Jahn-Teller distortion in the Mn^{3+} . For iron hexacyanoferrate a tetragonal transition is not expected and is not observed. There is some previous ex situ XRD showing this (Energy Environ. Sci., 2014, 7, 1643).

The three phase transitions for our PB-S3 sample were confirmed in our manuscript with *in-situ* synchrotron PXRD. The selected Le Bail fitted PXRD patterns from pristine, charge to 3.2V and 4.0V states are shown in Figure S8 c, d and e, respectively, which represents rhombohedral, cubic and tetragonal structure, respectively. The tetragonal phase at fully charged state 4.0V was confirmed and shows lattice parameter $a=10.1926 \text{ \AA}$ and $c=10.2085 \text{ \AA}$. The simulated PXRD results for three different phases are shown in Figure 4b, which is in consistent with our in-situ battery PXRD data. Theoretically, if PBA is fully charge to 4.0V, Fe^{2+} should be oxidized to Fe^{3+} , $[\text{Fe}(\text{CN})_6]^{3-}$ (the ferricyanide anion) which contains Fe^{3+} with a nominally $3d^5$ configuration. Ferricyanide is an octahedral molecular structure which can have a Jahn-Teller distortion. Therefore, we do not agree with Reviewer1's comment on it. In addition, the Reviewer 1 mentioned this reference "Energy Environ. Sci., 2014, 7, 1643", but this reference is totally different from our work as they reported a PBA with a sodium-poor cubic phase, with $\text{Na}_{0.61}$ only. In their electrochemical test part, the coin cell was discharge first, which means that sodium ions from sodium anode site insert into PBA

cathode first and the discharge capacity is 130 mAh g⁻¹ for the first cycle, this value is absolutely not the real capacity of their sample. According to the theoretical capacity 170 mAh g⁻¹ of sodium iron hexacyanoferrate, it can be calculated that if they charge the coin cell first, the initial discharge capacity would be lower than 40 mAh g⁻¹, which is much lower than capacity of our sample. The second discharge capacity of their sample goes up to 180 mAh g⁻¹, which is due to extra sodium-ions intercalated into PBA in the first cycle and provide more capacity. However, 180 mAh g⁻¹ is even higher than theoretical capacity (170 mAh g⁻¹) due to their weight error. Therefore, the capacity comparison between this reference and our work is not under the same condition. In terms of their ex-situ PXRD result, their coin cell was discharged first as well, which could not be used to compare with our in-situ result as our sample was charged first and then discharge. In common sense, the cathode materials provide sodium-ions in the whole full battery system and it should be charged first to provide sodium-ions and then discharged. Also, the accuracy of the ex-situ technique needs to be taken into consideration as the peak shift is minor, ex-situ PXRD technique could not get the stable voltage and hence the ex-situ results is not reliable than our synchrotron in-situ battery PXRD test.

Figure S8. (a) Synchrotron PXRD of PB-S3 sample at different voltage, (b) simulated PXRD of three different phases, (c-e) Le Bail fitted PXRD patterns from pristine PB-S3 (rhombohedral), charge to 3.2V (cubic) and 4.0V (tetragonal).

Can the authors explain how the Rietveld refinement was done? Was the occupancy of Fe₂C₄N allowed to relax and what was the treatment of water within the sample (about 16% by mass from TGA)?

All PXRD data analysis was done in TOPAS 5 software. The PXRD data was first indexed to get unit cell, lattice parameters, and crystal symmetry information. Then the indexed unit cell was used for Le Bail fitting the PXRD data to derive the suitable peak profile, and lattice parameters. These

derived data were fixed and used for further Rietveld refinement. The Rietveld refinement was done with initial structure models from ICDD PDF-4 2019 database. The occupancy of all atoms is allowed to refine to check the atomic occupancy with fixed APD values at $B = 1$. E.g. in PB-S1 sample in Table S1., the free refined CN ligand is $= 1.081$. As the results, we assumed and fixed the Fe and CN occupancy to be 100%. The structural water content may be disordered or without long range order. Another indication is if water is periodic located in the structure, space group $Fm-3m$ maybe reduced. Na positions was calculated by difference electron maps, then Na occupancy was refined. Please note that it's hard to locate water positions due to no periodic ordering of water positions across the unit cells (none constructive diffraction signal). Further work is under scope with low temperature neutron diffraction (at 4 Kelvin) in the future.

First cycle coulombic efficiency cannot be used for comparison as the material is oxidized by varying degrees.

The oxidization of Fe^{2+} is inevitable during synthesis process, however, the initial coulombic efficiency of our PB-S3 sample is still higher than almost all other reported co-precipitated $Na_2-xFeFe(CN)_6$ samples as shown in Table S4 in the Supporting information, as our efforts in protection material from oxidization. A lot of references reported that ICE of $Na_2-xFeFe(CN)_6$ is higher than 100%, indicating the amount of vacancies are high in their sample or the oxidation of their sample has not been controlled carefully.

Why do the authors think the specific capacity is lower than both the theoretical value and other values reported for iron hexacyanoferrate? The material is not better performing compared to others reported in terms of specific capacity and capacity retention.

The theoretical value of PBA is 170 mAh g^{-1} , however, it is difficult to achieve as the fast precipitation speed would cause vacancies with coordinated water, the specific capacity of our sample is lower than theoretical capacity of PBA is because the sodium content is not as full as 2.0 and there are vacancies and crystalline water in structure, also we limited charge-cut off voltage at 4.0 V in order to obtain stable cycling performance. However, we do not think the specific capacity of our PB-S3 sample is lower than other reported co-precipitated iron hexacyanoferrate samples, as shown in Table S4, because all references show capacity around $100 \sim 120 \text{ mAh g}^{-1}$, however, the cycle performance of our sample is better than most of co-precipitated $Na_2-xFeFe(CN)_6$ samples due to the high sodium content and reversible rhombohedral structure of our sample. As we mentioned before, although the hydrothermal method could fabricate PBA with higher capacity about 160 mAh g^{-1} , but it is not suitable for large-scale practical application and the co-precipitation is the only suitable method for mass production. The reference "Energy Environ. Sci., 2014, 7, 1643" reported 130 mAh g^{-1} and 180 mAh g^{-1} discharge capacity for the initial and second cycle, which is due to their sodium poor $Na_{0.61}FeFe(CN)_6$ material was discharged first in the initial cycle, sodium ions from sodium metal anode site inserted into cathode site and increased its capacity, which could not represent the real capacity of their material if the battery was charged first and followed by discharge. Obviously, the real capacity of their sample would be much lower than our PB-S3 sample capacity as their sample contains much lesser sodium content.

Reviewer #2 (Remarks to the Author):

This manuscript reported a scalable synthesis method of sodium-rich Fe-based Prussian white for sodium ion storage. The nucleation and precipitation process are presented. Moreover, the phase transition mechanism during the sodium storage process is also studied. However, as the authors mentioned in the introduction part, the as-proposed cathode has been reported elsewhere, which also demonstrated excellent cycling and rate performance. There is no breakthrough in terms of electrochemical performance in this cathode. Besides, the sophisticated phase transition in this material does not benefit long-term cycling stability in practical application. On the other hand, the co-precipitation method is widely used in the preparation of PBAs. Hence, the novelty of this paper is not high enough, especially for Nature Communications. It is more suitable for a specific energy storage related journal. I cannot recommend its publication. There are some other questions for the authors' attention.

Thanks for the comments from Reviewers #2, although the as-proposed cathode has been reported elsewhere and shows the excellent capacity and cycling performance, but we mentioned in introduction part that the synthesis method they used is hydrothermal method, which could not be used in real application as low-yield and NaCN might exist as by-product. However, in our work, we used a simple scalable mild co-precipitation method to fabricate $\text{Na}_{2-x}\text{FeFe}(\text{CN})_6$. The problem for co-precipitation method so far in other reference is that most of them demonstrate sodium-poor cubic phase with unstable cycling performance, by carefully controlling precipitation process, the cycling performance of our rhombohedral PB-S3 sample is better than most of reported $\text{Na}_{2-x}\text{FeFe}(\text{CN})_6$, as shown in Table S4 in supporting information, and also our sample has no any further doping or coating modifications, which is considered as a low-cost and easy-prepared cathode material for "real industry" materials for sodium-ion battery. A large-scale synthesis by using 100 L reactor is illustrated in Figure 5 and pouch full cell was fabricated to demonstrate the feasibility of practical application of our material. The new findings of highly reversible three phase transitions of our rhombohedral PB-S3 sample was found for the first time, which is due to rhombohedral $\text{Na}_{2-x}\text{FeFe}(\text{CN})_6$ contains higher sodium content than cubic phase, the synchrotron in-situ PXRD technique provides a mechanism for rhombohedral $\text{Na}_{2-x}\text{FeFe}(\text{CN})_6$ on sodium storage performance. The co-precipitation method is a common and well-known scalable synthesis method, but a systematically study with controlled particle size and morphology is demonstrated in our manuscript for the first time, which is extremely important for its scale-up production as most references exhibits nanosized particle materials, indicating the precipitation process has not been carefully controlled and the nanometer particle would make it difficult in manufacturing process. Benefiting from our controlled co-precipitation method, a scale-up synthesis of $\text{Na}_{2-x}\text{FeFe}(\text{CN})_6$ in 100 L reactor and pouch full cell with over a thousand times cycle life is presented in Figure 5 in manuscript. The breakthrough and novelty of our work are obvious, including nucleation process, phase transitions of rhombohedral $\text{Na}_{2-x}\text{FeFe}(\text{CN})_6$, large-scale fabrication and pouch full cell performance. None of them can be found in other reports so far.

1. Why do the authors choose Sample 1 and Sample 3 to compare? How's the electrochemical performance of Sample 2 and Sample 4?

We chose sample 1 and 3 as electrochemical investigated targets as they represent cubic and rhombohedral structure, respectively. Sample 1 demonstrates a sodium-poor cubic structure and

Sample 2, 3 and 4 are all rhombohedral phases, but sample 3 shows the strongest intensity from XRD results and the sodium content is the highest among them, therefore we selected sample 1 and sample 3 for further electrochemical performance study. Electrochemical performances of Sample 2 and 4 are shown as below, as they not performed better than sample 3 content, hence we did not put those data in manuscript.

2. Since the authors claimed that the ICE of the PB-S3 sample is beneficial for making full cell, it is more convinicible to configurate full cell to prove this.

The pouch full cell performance was added in manuscript as shown in Figure 5. The cathode materials in pouch cell was obtained from a scale-up production from 100 L reactor and hard carbon was used as anode material.

Figure 5 (a) Digital image of synthesis of Prussian white $\text{Na}_{2-x}\text{FeFe}(\text{CN})_6$ in 100 L reactor and powder of final product, (b) digital images of pouch full cell connected with red LED lights, (c) working mechanism of pouch full cell, (d) charge-discharge curves of pouch full cell, (e) cycling performance of pouch full cell.

3. How's the rate performance beyond 500 mA g⁻¹? The open framework of PBA is supposed to allow fast ion transportation.

The rate performance under current densities of 1000 and 2000 mA g⁻¹ was provided in Figure 3 in manuscript as below:

4. There is no significant difference in the diffusion coefficient (Figure 3h), making the statement “which could explain the superior cycling and rate performance of the high sodium content rhombohedral structure” a little weak. Moreover, the cycling stability is not only related to the kinetic properties. The discussion here is too general.

Thanks for your comments here, we have modified those descriptions in manuscript in red color, the minor difference between diffusion properties of cubic PB-S1 and rhombohedral PB-S3 sample is due to they both possess open framework structure, which is favourable for sodium-ions transportation. However, the sodium diffusion coefficient of rhombohedral PB-S3 sample is slightly higher than that of cubic PB-S1 sample, which can be ascribed to lower coordinated water content in structure. The better cycling and rate performance of PB-S3 sample is due to its highly reversible structure rhombohedral, the structure is relatively stable than irreversible cubic phase from our synchrotron in-situ battery PXRD results, as shown in Figure 4 and Figure S7.

5. The three-phase structural evolution should be further elucidated by Rietveld refinement.

The three phase transitions for our PB-S3 sample were confirmed in our manuscript, selected Le Bail fitted PXRD patterns from pristine, charge to 3.2V and 4.0V states are shown in Figure S8 c, d and e, respectively, which represents rhombohedral, cubic and tetragonal structure, respectively.

Figure S8. (c-e) Le Bail fitted PXRD patterns from pristine PB-S3 (rhombohedral), charge to 3.2V (cubic) and 4.0V (tetragonal).

Reviewer #3 (Remarks to the Author):

In this paper, the authors investigated $\text{Na}_2\text{-xFeFe}(\text{CN})_6$ Prussian blue analogues as a kind of low-cost cathode material in sodium-ion battery. The precipitation process during synthesis was systematically studied and some interesting results were found, including particle size, morphology and the relationship to structure and battery performance. The rate and cycling performance have been improved by obtaining the high crystalline microcube particle with sodium-rich rhombohedral phase. In-situ XRD analysis shows a reversible three phase transitions for rhombohedral $\text{Na}_2\text{-xFeFe}(\text{CN})_6$, which seems like a new mechanism of structural evolution for this material and has not been found in any other reports so far. The results are interesting and attractive which should be published here. But some minor revisions and questions should be addressed in the manuscript as well.

Synthesis methods were reported in other papers by using sodium-citrate as chelating agent like J. Mater. Chem. A, 2016, 4, 6036; Nano Energy 39 (2017) 273–283, what is the advantage between author's synthesis and others work?

Sodium citrate was used in co-precipitation synthesis of our work as it acted as chelating agent and sodium supplement at same time. The difference of our work with other reports is not only the influence of sodium citrate, as many other conditions would affect the precipitation process, like atmosphere and chelating time, as well as the amount of sodium citrate and the source of sodium salt were investigated. The unique part of our synthesis is adding the sodium citrate in $\text{Na}_4\text{Fe}(\text{CN})_6$ solution, which is important for the formation of cubic morphology as a slower reaction speed could be obtained, a clear comparison with only sodium citrate in FeSO_4 solution and without sodium citrate in $\text{Na}_4\text{Fe}(\text{CN})_6$ solution was demonstrated in Figure S2, which shows much smaller primary particles, indicating that the nucleation speed is still fast. Moreover, there are no references reporting the effect of sodium citrate in $\text{Na}_4\text{Fe}(\text{CN})_6$ solution yet, which is actually important to the particle size and morphology of $\text{Na}_2\text{-xFeFe}(\text{CN})_6$.

The evolution of morphologies of four samples in Fig. 2 is needed to be explained, why would the primary particles aggregate and the size of PB-S4 sample become smaller than that of PB-S3 with increasing sodium citrate concentration.

The amount of sodium citrate plays an important role in morphology and particle size of $\text{Na}_2\text{xFeFe}(\text{CN})_6$ material. However, when sodium concentration in reaction solution is not enough, the nucleation speed was still fast that aggregated small primary particle was observed, and the primary particle is hard to grow bigger. When sodium content is too high, a uniform cubic morphology could be obtained by a little smaller particle sized was found, which is due to supersaturation of sodium citrate in reaction solution.

In the GITT test in electrochemical performance, the “ τ ” is missing from the calculation formula of D_{Na^+} in the Supporting Information, the GITT results needed to be verified.

Thanks for reminding, we are sorry for the mistake on GITT calculation formula, the “ τ ” was added in manuscript, the D_{Na^+} was calculated by using formal containing “ τ ”.

There is a difference between unit cell volume of rhombohedral PB-S3 sample in Figure 4 and its XRD refined volume, could author explain it.

The primitive unit cell of rhombohedral structure could be identically transformed from hexagonal structural setting to rhombohedral setting, the volume for rhombohedral phase in Figure 4 is divided by three from hexagonal setting, which is provided in Table S2 in the supporting information.

There should be more explanation for XRD stimulation in different three phases in Figure S8 in the supporting information. How did author confirm the tetragonal phase was formed.

The three phase transitions for our PB-S3 sample were confirmed again in our manuscript, selected Le Bail fitted PXRD patterns from pristine, charge to 3.2V and 4.0V states are shown in Figure S8 c, d and e, respectively, which represents rhombohedral, cubic and tetragonal structure, respectively. The tetragonal phase at fully charged state 4.0V was confirmed and shows lattice parameter $a=10.1926 \text{ \AA}$ and $c=10.2085 \text{ \AA}$. The simulated PXRD results for three different phases are shown in Figure 4b, which is in consistent with our in-situ battery data.

Figure S8. (a) Synchrotron PXRD of PB-S3 sample at different voltage, (b) simulated PXRD of three different phases, (c-e) Le Bail fitted PXRD patterns from pristine PB-S3 (rhombohedral), charge to 3.2V (cubic) and 4.0V (tetragonal).

Reviewers' comments:

Reviewer #1 (Remarks to the Author):

The performance of the material is comparable to reported materials synthesized under similar conditions, Table S4, but inferior (specific capacity, cycle life) than those synthesized through the hydrothermal route. We do appreciate arguments for a scalable synthesis route and a comparison to other coprecipitation synthesized Prussian White materials is justified. Nevertheless, the performance in terms of specific capacity and cycle life is only comparable and not much improved. Additionally, using chelating agents is also not a novel synthesis method. We feel that, although the work is well thought through with thorough characterisation, the performance of the material is not exceptional and there is not sufficient novelty for publication in Nature Communications.

There remain specific issues with the interpretation of the in situ XRD. Figure S8c d and e show diffractograms that are visually indistinguishable. There are characteristic peak splitting's that correspond to distortions away from cubic to either rhombohedral or tetragonal. These diffractograms show none of these. The fact that the rhombohedral and tetragonal phases can be fitted in the Le Bail refinement is because they are similar but lower symmetry phases compared to the cubic. If the cubic phase can be fitted with equivalent residuals, then it should be concluded that the phase is cubic. From previous literature the rhombohedral phase is expected in the discharged state and in S8e there is evidence of the peak at 10.80 splitting. We are unsure why S8e does not show this peak splitting. We are more concerned in the conclusion that in the charged state the material is tetragonal. There is no peak splitting characteristic of tetragonal distortion and the reported lattice parameters ($a=10.1926 \text{ \AA}$ and $c=10.2085 \text{ \AA}$) differ by only 0.16%. We believe that this is a cubic phase being fitted with an unnecessarily low symmetry tetragonal phase. Could the authors refine this pattern in the same manner with the cubic phase and report a comparison of the residuals?

Reviewer #2 (Remarks to the Author):

The authors have well addressed all the issues. The revision meets my concern and can be recommended the acceptance for publishing in Nature Communications.

Reviewer #3 (Remarks to the Author):

According to the previous comments, some modifications have been made for revised manuscript by authors, including structural refinement for three different phases from in-situ synchrotron PXRD data, pouch full cell performance, and more detailed descriptions of experimental results. The quality of the paper has been improved dramatically and I think acceptance could be considered by reasons below.

The nucleation and particle growth process for sodium-rich rhombohedral $\text{Na}_{2-x}\text{FeFe}(\text{CN})_6$ material are investigated, which is significant for understanding the relationship between particle size, morphology, and electrochemical performance of Prussian blue analogues for sodium-ion batteries. The controlled co-precipitation method could fabricate high quality PBAs materials with uniform microcube morphology, high sodium content and superior electrochemical performance.

The simple and scalable co-precipitation method for synthesizing sodium-rich rhombohedral $\text{Na}_{2-x}\text{FeFe}(\text{CN})_6$ provided by authors is with potential to realize practical application. A large-scale fabrication of rhombohedral $\text{Na}_{2-x}\text{FeFe}(\text{CN})_6$ in a 100 L reactor with kilograms yield is demonstrated,

and a pouch full cell shows stable cycling performance over 1000 times. It is a big breakthrough for scale-up synthesizing PBAs cathode materials for sodium ion batteries, the long-term cycling performance of full cell is outstanding as well.

The highly reversible three phase transitions (rhombohedral-cubic-tetragonal) for sodium-rich rhombohedral $\text{Na}_{2-x}\text{FeFe}(\text{CN})_6$ material for sodium ion battery is first reported, which reveals the intrinsic structural evolutions during cycling.

There remain specific issues with the interpretation of the in situ XRD. Figure S8c d and e show diffractograms that are visually indistinguishable. There are characteristic peak splitting's that correspond to distortions away from cubic to either rhombohedral or tetragonal. These diffractograms show none of these. The fact that the rhombohedral and tetragonal phases can be fitted in the Le Bail refinement is because they are similar but lower symmetry phases compared to the cubic. If the cubic phase can be fitted with equivalent residuals, then it should be concluded that the phase is cubic. From previous literature the rhombohedral phase is expected in the discharged state and in S8e there is evidence of the peak at 10.80 splitting. We are unsure why S8e does not show this peak splitting. We are more concerned in the conclusion that in the charged state the material is tetragonal. There is no peak splitting characteristic of tetragonal distortion and the reported lattice parameters ($a=10.1926 \text{ \AA}$ and $c=10.2085 \text{ \AA}$) differ by only 0.16%. We believe that this is a cubic phase being fitted with an unnecessarily low symmetry tetragonal phase. Could the authors refine this pattern in the same manner with the cubic phase and report a comparison of the residuals?

Answer: Thanks for the comments. We would like to disagree with the comment of "Figure S8c d and e show diffractograms that are visually indistinguishable." (Now changed to Figure 5d e and f) We draw the conclusion of highly reversible three phase transformations based on systematic Le Bail and/or Rietveld refinement of all datasets, not only on 'visually distinguishable'. Based on our experimental data and calculated PXRD pattern in Figure 5 a, b, only rhombohedral phase should show the splitting of the (110) and (104) reflections, while cubic and tetragonal phases should show no peak splitting in the displayed range of PXRD patterns.

In addition, if we carefully check PXRD peak positions shifting trend in Figure 5a, b. E.g. near the latter stage of the charge process ($> 3.0 \text{ V}$), the PXRD peaks change from shifting from lower to higher angle, to shift from higher to lower angle. (before we discharge the battery) This can NOT be explained by a single rhombohedral phase. This clearly indicates the phase transformation during this process.

Figure 5. (a) Synchrotron PXRD of PB-S3 sample at different voltage, (b) simulated PXRD of three different phases, (c) PXRD patterns of electrode of pristine and discharged to 2.0V state, (d-e) Le Bail fitted PXRD patterns from pristine PB-S3 (rhombohedral), charge to 3.2V (cubic) and 4.0V (tetragonal).

Figure 5c was added in manuscript, which shows the PXRD patterns of electrode of pristine and discharged to 2.0V state from laboratory XRD facility. Obviously, the split peaks of (110)/(104) of rhombohedral phase (around 24°) could be observed, the broadening of split peaks in synchrotron PXRD data might be caused by the strong background of Al.

Figure 4. Investigation of phase transitions of $\text{Na}_{1.73}\text{Fe}[\text{Fe}(\text{CN})_6]\cdot 3.8\text{H}_2\text{O}$ sample with rhombohedral structure during cycling. (a) 2D contour plot of (012) reflection, (b) normalized volume during charge-discharge process obtained from synchrotron in-situ PXRD patterns of rhombohedral PB-S3, (c) (012), (110)/(104) and (024) reflections of synchrotron PXRD patterns

By combining high resolution synchrotron PXRD technique and parametric refinement method (Ref. <https://doi.org/10.1107/S0021889806043275>), we are comfortable to get lattice parameters resolution down to 0.002 Å range. This is enough to distinguish the structural distortion such as lattice parameters ($a=10.1926$ Å and $c=10.2085$ Å). (BTW: 0.16% is not a very small distortion). We understand the principal to fit the highest symmetry/reasonable structural model as possible. We have compared the fitting of a tetragonal phase PXRD pattern with a tetragonal structural model and cubic structural model, with $R_{wp} = 3.45\%$ and 4.56% , respectively.

Overall, all these results prove our conclusions of three phase transitions are defensible.

REVIEWERS' COMMENTS:

Reviewer #4 (Remarks to the Author):

Dear authors,

I have been asked to give one more opinion on the in situ XRD part of your manuscript.

In the current state of the figure, I agree with the reviewer about the in situ XRD part. I do not think that, in its present form, the figure allows to distinguish between cubic and rhombohedral phases. It would be easy to mistake a peak broadening for a peak splitting in these conditions.

I recommend to plot the figure by narrowing the peaks and expanding both the Y axis and the space between consecutive XRD patterns.

with Best Regards,

Reviewer 4#:

In the current state of the figure, I agree with the reviewer about the in situ XRD part. I do not think that, in its present form, the figure allows to distinguish between cubic and rhombohedral phases. It would be easy to mistake a peak broadening for a peak splitting in these conditions.

I recommend to plot the figure by narrowing the peaks and expanding both the Y axis and the space between consecutive XRD patterns.

Answer: Thank you very much for your advice, the peak broadening of (110)/(104) in Fig. 4 c might be caused by a slightly changed cell orientation and internal strain of electrodes during the *operando* measurement. As you suggested, we narrowed the peaks of (110)/(104) between 10.5~11.3 degrees and expanded the Y axis and the space as well, as shown in Fig. 4 c below, however it is still hard to see peak splitting around 11 degree as the low intensity from small amount of sample in the cell cathode. We draw the phase transformations conclusion not based on a single peak analysis, but on the whole pattern fittings of first few diffraction peaks (Fig. 5 d-f). More detailed structure information of PB-S3 sample are demonstrated in Fig. 5, as shown below. The double peaks of (110)/(104) around 24 degree as a feature of rhombohedral phase could be clearly observed in Fig. 5 c and the structure was maintained during first two cycles. Combing these results in Fig. 5 with *in-situ* PXRD data in Fig. 4, we could draw conclusion of three reversible phase transitions for our PB-S3 sample.

Fig. 4 Investigation of phase transitions of $\text{Na}_{1.73}\text{Fe}[\text{Fe}(\text{CN})_6] \cdot 3.8\text{H}_2\text{O}$ sample with rhombohedral structure during cycling. **(a)** 2D contour plot of (012) reflection, **(b)** normalized volume during charge-discharge process obtained from synchrotron in-situ PXRD patterns of rhombohedral PB-S3, **(c)** (012), (110)/(104) and (024) reflections of synchrotron PXRD patterns, **(d)** schematic phase evolutions during cycling of rhombohedral PB-S3.

Fig. 5 Structural characterization of PB-S3 sample during cycling. **(a)** Synchrotron PXRD of PB-S3 sample at different voltage, **(b)** simulated PXRD of three different phases, **(c)** PXRD patterns of electrode of pristine and discharged to 2.0 V state, **(d-f)** Le Bail fitted PXRD patterns from pristine PB-S3 (rhombohedral), charge to 3.2 V (cubic) and 4.0 V (tetragonal).